# Relationship between the Antioxidant Activity and Allelopathic Activities of 55 Chinese Pharmaceutical Plants

**DOI:** 10.3390/plants11192481

**Published:** 2022-09-22

**Authors:** Yoshihiro Nomura, Kwame Sarpong Appiah, Yoko Suzuki, Yoshiharu Fujii, Qile Xia

**Affiliations:** 1State Key Laboratory for Managing Biotic and Chemical Threats to the Quality and Safety of Agro-Products, Zhejiang Provincial Key Laboratory of Fruit and Vegetables Postharvest and Processing Technology, Ministry of Agriculture and Rural Affairs Key Laboratory of Post-Harvest Handling of Fruits, Institute of Food Science, Zhejiang Academy of Agricultural Sciences, Hangzhou 310021, China; 2United Graduate School of Agriculture, Tokyo University of Agriculture and Technology, Tokyo 183-8509, Japan; 3Department of Crop Science, University of Ghana, Legon, Accra P.O. Box LG 44, Ghana; 4Department of Sustainable Production, Institute of Xinjiang Environmental Protection, Urumqi 830000, China; 5Aromatic Repos, AHOLA, A2 Soleil Jiyugaoka, Meguro, Tokyo 152-0035, Japan

**Keywords:** antioxidant activity, allelopathic activity, dishpack method, ORAC, DPPH-RSA, total phenolics, sandwich method

## Abstract

Pharmaceutical plants contain several phytochemicals that are sources of myriad biological activities. These biological activities can be explored in multiple fields for the benefit of mankind. Pharmaceutical plants with high ethnobotanical indices (i.e., use value and relative frequency of citation) were reported with the potential to inhibit lettuce elongation through leachates and volatiles. The focus of the study was to assess Chinese pharmaceutical plants for both antioxidants, as well as allelopathic potentials to explore any underlying relationship. The estimation of antioxidative capacity and content of total phenolics (TPC) for the 55 Chinese pharmaceutical plants was conducted by the assays of DPPH radical scavenging activity (DPPH-RSA), oxygen radical absorbance capacity (ORAC) and the means of Folin–Ciocalteu. The estimation of the activity of allelopathy for collected medicinal plants was done by adopting the sandwich method for plant leachates and the dishpack method for volatile constituents, respectively. The fruits of sea buckthorn (*Hippophae rhamnoides*) had the most remarkable ORAC value (168 ± 7.04 μmol TE/g) and DPPH radical scavenging activity (440 ± 7.32 μmol TE/g) and contained the highest contents of total phenolic compounds (236 ± 7.62 mg GAE/g) in the 55 pharmaceutical plant species according to the results. In addition, sea buckthorn showed dominant allelopathic potential through plant leachates evaluated by using the sandwich method. Star anise (*Illicium verum* Hook. f.) showed conspicuous allelopathic activity through plant volatiles assessed by the dishpack bioassay method. Among the same plant species, antioxidative ability and total phenolics, in comparison with potential allelopathy of medicinal herbs indicated that volatile allelochemical had a weak active effect (r = 0.407 to 0.472, *p* < 0.01), with antioxidant capacity by the dishpack method. However, the evaluation by the sandwich method showed a significant positive correlation (r = 0.718 to 0.809, *p* < 0.001) with antioxidant capacity. Based on these results, a new hypothesis is that the antioxidant activity of plants may have an involvement with the potential allelopathic activity.

## 1. Introduction

Many active phytochemicals are derived from pharmaceutical plants, and most of these compounds are beneficial to human health and the ecosystem. Bioactive compounds in plants are responsible for the antioxidative effect, allelopathic activity and other biological effects of plant species [1,2,3]. The human body produces free radicals and these free radicals can start chain reactions, which can cause damage or death to the cell when the chain reactions occur in the cell [1,3]. Antioxidants are mainly responsible for the body’s defense mechanism related to free radicals. Thus, the intake of plant-derived antioxidants can help mitigate degenerative diseases caused by oxidative stress, such as cancer, Parkinson’s, Alzheimer’s or atherosclerosis [4,5]. Allelopathy is a phenomenon involving either direct or indirect and either beneficial or adverse effects of a plant (including microorganisms) on other plant species through the release of allelochemicals in the environment [2]. These allelochemicals are produced by allelopathic organisms and are mainly released into the environment through leaching, secretion of roots or decomposition by microorganisms [6].

In recent years, several studies have focused on medicinal plants for their antioxidant and phytotoxic potential. Natural antioxidants, such as phenols in various medicinal plants, can play an antioxidant role by decomposing peroxides, absorbing free radicals in the human body, and quenching oxygen ions in singlet or triplet states [7]. Some Indian traditional herbs, such as myrobalan, Indian almond and Emblica, were reported with strong antioxidant activities, with a high amount of phenolic compounds [8]. Studies on the allelopathic activity among medicinal plants provide a good alternative to diversifying weed control techniques. Macaroon oil and its flavonoids can inhibit the growth of most weeds at a certain dose and could maintain stability for about 7 days [9]. The Triones isolated from Sargassum are phytotoxins, which have been shown to weaken the emergence of corn weeds [10]. Trione is a plant growth inhibitor isolated from Sargassum, which can control glyphosate-tolerant weeds effectively [11]. Three medicinal herbs, including rhubarb, *Saussurea involucrata,* and *Potentilla discolor*, were reported with potential plant growth inhibitory effects on some traditional agriculture crops [12]. China is famous for its wide land and abundant pharmaceutical plant resources. According to a recent survey, there are 10,608 species of higher pharmaceutical plants in China, accounting for 83.4% of all its medicinal biological resources [13]. In the Labiatae, Leguminosae and Compositae of traditional Chinese medicinal plants, a large number of secondary metabolites formed by phenylpropanoids, including flavonoids, monophenols, lignans, phenolic acids and other phenolic substances, have been found [14]. They play important roles in anti-oxidation, free radical scavenging and anti-inflammatory measures [15]. In the *Rumex japonicus* and *Rehmannia glutinosa*, two kinds of famous Chinese herbal medicines, 2-6-dibutylphenol, protocatechuic acid, ferulic acid, lauric acid and ferulic acid and other phenolic allelopathic substances were identified, which were proved to significantly reduce the seedling growth of lettuce, barley grass and sesame. In addition, the important role of phenolic substances from other pharmaceutical plants in allelopathy was also largely confirmed [16]. Although Appiah et al. [17] reported a significant correlation between ethnobotanical indices (use value and relative frequency of citation) and allelopathy through leachates from medicinal plants, there is no scientific study on the relation between the allelopathy and antioxidant activity of Chinese pharmaceutical plants and the study will be of great significance and reference value. Understanding the relevance between allelopathy and antioxidation will help us to screen a large number of raw materials for allelopathic potential efficiently. Given the dual role of phenolic substances in allelopathy and antioxidation, there was a hypothesis that antioxidative capacity may have an association with the allelopathic effect. Plant secondary metabolites, including phenolic compounds or other antioxidative compounds through leaching, are released into organisms and may cause the accumulation of soluble biochemicals and the phenomenon of allelopathy. To test this hypothesis, 55 Chinese pharmaceutical plant species were assessed for antioxidative activity and phenols content, conducted by analysis of oxygen radical absorbance capacity (ORAC), DPPH radical scavenging activity (DPPH-RSA) and Folin–Ciocalteu. The evaluation of the allelopathy of these plant species was applied using the sandwich method and the dishpack method. 

## 2. Results

### 2.1. Antioxidative Capacity and Total Phenolics Content of the 55 Chinese Pharmaceutical Plants

To determine the antioxidant capacities of the 55 Chinese pharmaceutical plants collected, experiments of ORAC (oxygen radical absorbance capacity) and DPPH-RSA (DPPH radical scavenging activity) were conducted. The content of total phenolics (TPC) of the 55 plant samples was determined by the Folin–Ciocalteu method. The antioxidative ability and TPC are shown in Table 1. The ORAC values varied from 56.3 ± 0.120 μmol TE/g to 168 ± 7.04 μmol TE/g. The top Chinese plant species, which had relatively high ORAC values in this study were the fruits of *Hippophae rhamnoides* (168 ± 7.04 μmol TE/g) and leaves of *Hippophae rhamnoides* (166 ± 6.05 μmol TE/g). This was followed by *Rosa multiflora* (159 ± 5.38 μmol TE/g), *Punica granatum* (149 ± 5.62 μmol TE/g), and *Ribes nigrum* (148 ± 5.23 μmol TE/g). The DPPH radical scavenging activity (DPPH-RSA) showed wide variation and ranged from 71.3 ± 0.55 μmol TE/g to 440 ± 7.32 μmol TE/g. Among the top Chinese plant species that showed relatively high DPPH-RSA values were the leaves of *Hippophae rhamnoides* (440 ± 7.32 μmol TE/g) and *Rosa multiflora* (426 ± 7.02 μmol TE/g). This was followed by the fruits of *Hippophae rhamnoides* (420 ± 7.15 μmol TE/g), *Punica granatum* (399 ± 6.40 μmol TE/g), and *Lycium ruthenicum* (387 ± 5.21 μmol TE/g). TPC for the evaluated medicinal species was in the range of 56.3 ± 0.260 mg GAE/g to 236 ± 7.62 mg GAE/g.

The fruits of *Hippophae rhamnoides* (236 ± 7.62 mg GAE/g), leaves of *Hippophae rhamnoides* (232 ± 7.40 mg GAE/g), *Rosa multiflora* (224 ± 7.51 mg GAE/g), *Punica granatum* (221 ± 6.48 mg GAE/g), and *Ribes nigrum* (217 ± 7.33 mg GAE/g) were among the species with high phenolic acid content. Sea buckthorn (*Hippophae rhamnoides*) showed the highest antioxidative activity at ORAC and DPPH-RSA determination, as well as dominant phenolic contents. Sea buckthorn was extracted by water extraction, soxhlet extraction and impregnation, respectively, and its antioxidant value was 164.03, 133.31 and 86.35 Trolox equivalents (TE) per gram, respectively, while phenolics content was 60.22, 43.77 and 28.35 mg GAE/g mg/g, respectively [18]. In addition, the methanol extract of rosehip (*Rosa multiflora*) has been reported to have a significant ORAC value of about 1085 TE μmol/g [19]. Sepulveda et al. [20] reported that the ORAC value between 12.7 to 24.4 mmol/L was found in pomegranate (*Punica granatum*). Moyer et al. [21] tested 32 genotypes of Ribes and found antioxidant capacity in Ribes ranged from 17 to 116 µmol TE/g. Flavonoids and other phenolic substances exist in various parts of sea buckthorn. These compounds have the effects of cellular antioxidation and prevention of cancer caused by inflammation and gene mutation [22]. There is an interesting report that the leaves and fruits of sea buckthorn contain isorhamnetin, quercetin and kaempferol, while quercetin-3-galactoside had the highest content in the plant [23]. In addition, the remarkable antioxidative ability at 128 mg vitamin C equivalents/L was shown in the report of Heo et al. [24]. 

### 2.2. Evaluation of Allelopathic Potential from 55 Chinese Medicinal Plants

In this section, the potential of allelopathy of 55 Chinese pharmaceutical plants was evaluated. The sandwich and dishpack bioassays for testing the growth inhibitory effects of water-soluble leachates and the volatile allelochemicals, respectively, were adopted. The lettuce radicle growth after treatment with oven-dried plant samples using the sandwich and dishpack methods is shown in Table 2. The normal distribution from the radicle elongation of these plant species by the sandwich method is shown in Figure 1. The lettuce radicle and hypocotyl elongation was appraised by the “standard deviation value” (SDV). About 89.1% of medicinal plants suppressed the lettuce elongation, but six species stimulated the growth of lettuce in the sandwich bioassay. The radicle elongation of lettuce seedlings with 10 mg plant sample treatment was in the range of 7.30 ± 0.35–108 ± 10.1%. The lowest radicle elongation (7.30 ± 0.35% of untreated control) came from *Hippophae rhamnoides* (sea buckthorn fruit), followed by sea buckthorn leaf (10.0 ± 0.52% of untreated control) and the fruit of *Illicium verum* (14.0 ± 0.48%). Moreover, there was radicle elongation of 20–40% among 8 plants, 41–60% in 12 plants, 61–80% in 14 plants and 81–100% in 11 plants. According to standard deviation variance analysis, five plant species showed very strong inhibitory activity and were categorized as “***”. These species included *Hippophae rhamnoides* fruit (7.30 ± 0.35% of untreated control), *Hippophae rhamnoides* leaf (10.0 ± 0.52% of untreated control), *Illicium verum* (14.0 ± 0.48% of untreated control), *Gossypium herbaceum* (19.3 ± 0.21% of untreated control), *Ribes nigrum* (21.0 ± 0.70% of untreated control), and *Stachys geobombycis* (22.3 ± 0.65% of untreated control). Similarly, Bao et al. [25] reported that the fruits of *Hippophae rhamnoides* showed a high inhibitory effect on lettuce elongation among the 22 medicinal plants samples evaluated and the radicle growth elongation was just 4% of control. The antioxidant activity, antibacterial and notable anticancer potential in *Illicium verum* have been reported in a previous study [26]. Flavonoids are absent in *Gossypium herbaceum* and possess maximum antimicrobial activity [27]. On the contrary, six plant species showed stimulated allelopathic activity (i.e., radicle extension > 100% of untreated control) on lettuce. Plant samples, which promoted the growth of seedlings, included *Amaranthus viridis* (101 ± 9.23%), *Arnebia euchroma* (102 ± 9.64%), *Sinapis alba* (104 ± 9.70%), *Armeniaca vulgaris* (106 ± 10.0%), *Carya cathayensis* (107 ± 9.85%) and *Morus macroura* (108 ± 10.1%). Previous studies also had similar results. Yucca root (water celery) was confirmed to promote the growth of tested plants exceeding 20% [28]. 

The normal distribution of the radicle elongation of the 55 Chinese pharmaceutical plants with 250 mg sample treatment in the dishpack method is shown in Figure 2. The radicle growth of lettuce seedlings was in the range of 16.0 ± 0.14–112 ± 10.5% of the untreated control. The lowest radicle elongation, i.e., highest inhibitory effect (16.0 ± 0.14% of control) was found in fruits of star anise (*Illicium verum*). Similar to the results of this study, Kang et al. [29] observed the highest inhibitory effect on lettuce radicle and hypocotyl by the leaves of *I. verum*. Trans-anethole has been identified as the main component in the essential oil of star anise. Another study reported the functions of the antioxidant and antibacterial activities of the essential oil of star anise [30]. Lettuce radicle elongations of 20–40% (4 plants), 41–60% (9 plants), 61–80% (15 plants) and 81–100% (17 plant) was recorded. Based on the standard deviation variance analysis, plants with the highest volatile allelopathy included *Illicium verum* (16.0 ± 0.14% of untreated control), *Angelica sinensis* (29.1 ± 0.81% of untreated control), *Rheum officinale* (36.2 ± 1.13% of untreated control) and *Indigofera tinctoria* (38.0 ± 0.82% of untreated control) with lettuce radicle elongation of 16.0, 29.1, 36.2 and 38.0 (% of untreated control), respectively. On the contrary, ten plants showed stimulatory growth in lettuce. These species were *Senecio scandens* (101 ± 3.02% of untreated control), *Equisetum hyemale* (101 ± 6.45% of untreated control), *Armeniaca vulgaris* (101 ± 8.86% of untreated control), *Fritillaria walujewii* (102 ± 3.95% of untreated control), *Morus macroura* (104 ± 9.31% of untreated control), *Carya cathayensis* (105 ± 10.20% of untreated control), *Amaranthus viridis* (106 ± 9.63% of untreated control), *Cicer arietinum* (108 ± 9.22% to control), *Ziziphus jujuba* (110 ± 10.4% to control) and *Solanum nigrum* (112 ± 10.5% of untreated control). Volatile plant species have been found to have growth-promoting effects, especially on the growth of hypocotyls. Kang et al. [30] confirmed that the growth promotion rate of Kafir lime, and red-orange were in the range of 70–100%.

### 2.3. Relationship between Antioxidant Capacity and Allelopathic Activity

In this study, the antioxidant capacity and allelopathic activity of 55 Chinese pharmaceutical plants were evaluated their antioxidant capacity and allelopathic activity. To determine the relationship between antioxidant capacity and allelopathic activity, a correlation analysis was performed based on the results of antioxidative activity, total phenolics content, and lettuce growth inhibition evaluated by the sandwich method using the collected medicinal plant samples (Figure 3). Moreover, the correlation relationship between antioxidative activity, total phenolics content and lettuce growth inhibition evaluated by the dishpack method using Chinese medicinal plants was conducted (Figure 4). A significantly strong positive correlation (r = 0.791; *p* < 0.001, *n* = 55) between the ORAC values (antioxidant activity) and the plant growth inhibition by the sandwich method was observed (Figure 3a). Although significant, the positive correlation (r = 0.429; *p* < 0.01, *n* = 55) between the ORAC values and the plant growth inhibition by the dishpack method was weak (Figure 4a).

The relationship between the DPPH-RSA activity and inhibitory effect was shown in Figure 3b and Figure 4b. The DPPH-RSA activities in the plant species had a strong significant positive correlation (r = 0.809; *p* < 0.001, *n* = 55) with lettuce growth inhibition by sandwich method, while the growth inhibition observed by the dishpack method had a weak correlation (r = 0.472; *p* < 0.01, *n* = 55). Phenolic compounds play important roles in the allelopathic activities and antioxidative potential of plants. In this study, plants with high total phenolic content showed a strong growth inhibitory effect for both the sandwich method and dishpack method (Figure 3c and Figure 4c). The positive correlation association between content of total phenols and plant growth inhibitory activity by sandwich method (r = 0.718; *p* < 0.001, *n* = 55) was stronger than the dishpack method (r = 0.407; *p* < 0.01, *n* = 55). The results indicated that antioxidative capacity, which was evaluated by ORAC values, DPPH radical scavenging activity and total phenolics content had a stronger relationship with soluble bioactive allelochemicals than volatile allelochemicals. 

## 3. Discussion

In the study, 55 Chinese pharmaceutical plants were collected to evaluate their antioxidative capacity and allelopathic effects. Antioxidant capacity was defined as the level of ORAC and activity of DPPH radical scavenging, as well as total phenolics content. Allelopathy was described as the inhibitory or stimulatory effects of oven-dried plant samples on lettuce growth through plant leachates or plant volatiles. The results of the study showed that the fruits of sea buckthorn (*Hippophae rhamnoides*) had the highest allelopathic potential (through leachates) and also the highest antioxidative activity among the plant samples evaluated. Star anise (*Illicium verum*) showed allelopathic through volatile constituents and also had a strong antioxidant effect. In addition, allelopathic activity from soluble plant leachates had a stronger positive correlation with antioxidant capacity than the plant volatiles. A new assumption that the antioxidant activity of plants is associated with allelopathic potential through leachates is proposed. 

Generally, plant allelochemicals produced by secondary metabolism are released through root exudates, leachates, volatiles or decomposition of residues to affect other organisms [31]. It has been reported that phenolic compounds are important compounds involved with allelopathy and many of these compounds act as the main oxidative components to exert anti-inflammatory, anti-viral and other biological effects [32]. Krasowsk et al. [33] reported that phenolic compounds, including flavonoids and phenolic acid, are secondary plant metabolites that act as main antioxidants to exert anti-allergic, inflammation inhibition, treatment of diabetes and thrombotic, fungus and viral suppression. *Illicium verum* was reported with a strong antioxidative capacity according to previous research [26]. Our group reported that *Illicium verum* had the strongest activity on growth inhibition by volatile components and identified shikimic acid, a kind of phenolic acid as an allelochemical in the plant [34]. Golisz et al. [35] reported that rutin is the main allelochemical in polish buckwheat when tested by total activity, which was determined by the concentration and inhibitory effect of the compound. Rutin is a common phenolic compound in the berries of sea buckthorn with a high concentration [36]. Similar to the results of this study, other studies showed that *Hippophae rhamnoides* (sea buckthorn) has strong allelopathic potential and antioxidative capacity [23]. Due to the various biological activities, it could be inferred that phenolic compounds will be extensively used in ecological agriculture and the promotion of human health. On the other hand, as shown in Figure 3c and Figure 4c, although there is more of a positive influence on total phenol and allelopathy by sandwich method than the dishpack method, the levels of phenol compounds were not the only reason for antioxidant and allelopathy. Tyrosinase is another putative compound with both antioxidative activity and allelopathic effects. Nakajima et al. [37] found that tyrosinase inhibitory activity has a big contribution to the inhibition of lettuce growth. Tyrosinase, also called catechol oxidase, is a kind of enzyme, which contains copper. It exists in plant and animal tissues and catalyzes the formation of melanin and other pigments from tyrosine oxidation. Thus, the production of tyrosine is harmful to the amount of total phenol, as well as antioxidant capacity. The plant, which has a high activity of tyrosine, will also influence the allelopathic activity.

## 4. Materials and Methods

### 4.1. Samples and Experimental Resources

In the study, 55 Chinese pharmaceutical plants were collected by the Institute of Xinjiang Environmental Protection and Yunnan Medicinal Botanical Garden for the evaluation of their biological activities. After lyophilization at −20 °C for 24 h, plant materials were crushed into powder by a grinder and filtered with a 100-hole sieve. Seeds of Lactuca sativa were adopted as receptor plants and purchased from Takii Seed corporation (Takii Seed corporation, Japan, Kyoto) for the evaluation of allelopathic effects of the collected plant samples. As the standards, synthetic chemicals of gallic acid, Na_2_CO_3,_ Trolox (6-hydroxy-2,3,7,8-tetramethylchroman-2-carboxylic acid), AAPH (2,2-azobisdihydrochloride), K_2_HPO_4_, DPPH (1,1-diphenyl-2-picrylhydrazyl), fluorescein and Folin–Ciocalteu’s phenol reagent were obtained through Industries of Wako Pure Chemical (Wako Pure Chemical, Japan, Osaka). 

### 4.2. ORAC, DPPH-RSA and Folin-Ciocalteu Assay

Assay of oxygen free radical absorption capacity (ORAC) can prevent the degradation and loss of fluorescence of fluorescein probes by scavenging the peroxy groups related to AAPH (2,2-azodihydrochloride) [38]. The ORAC test of the 55 Chinese medicinal plant extracts was conducted based on Watanabe et al. [39] with a little modification. Two grams of powdered plants were dissolved in 40 mL of 80% methanol. The sample solution was ultrasonic-treated for 1 h and centrifuged for 15 min to obtain the plant extracts and diluted to 1 mg/mL. Trolox solution (control, 5 μM, 10 μM, 20 μM, 50 μM and 100 μM), fluorescein stock solution (1.2 mmol/L), and AAPH solution (500 μmol/L) were prepared by dissolving in a phosphate buffer (pH 7.4). The measurement was done by using 96-well black plates and a 96-well microplate reader (Infinite 200, Tecan, Japan, Tokyo). The Trolox solution (20 μL), plant crude extracts (20 μL) and phosphate solution (20 μL) were, respectively, added to suitable wells. Then, the fluorescein stock solution (200 μL) and AAPH solution (75 μL) were injected into the wells. Emission at 535 nm and excitation at 485 nm was set up on the fluorescence intensity. 

DPPH (1,1-diphenyl-2-pyridyl hydrazide) is widely used to evaluate the level of free radical elimination in natural foods due to its stability [40]. The DPPH test is based on the interaction between the free radicals of the stable *2*,2-diphenyl-1-pyridyl hydrazide and hydrogen ions by the method of Choi et al. [41]. In general, the stock solution of DPPH was prepared at 200 μM. Phosphate buffer of Trolox was obtained at 50 μM, 100 μM, 200 μM, 400 μM and 800 μM. Plant materials were dissolved in 80% methanol to obtain the plant extract. The measurement was done by using 96-well crystal plates (96 Flat Bottom Transparent) and a 96-well microplate reader (Infinite 200, Tecan, Japan, Tokyo). Plant extract (20 L), phosphate buffer (80 μL) and DPPH solution (100 μL) were mixed with suitable wells. The fluorescein was determined by adopting the microplate viewer at an absorbance of 517 nm. DPPH radical scavenging capacity was defined as Trolox equivalent (TE) (μmol) per gram of sample, based on the concentration of 50% inhibitory effect of DPPH staining. The DPPH radical scavenging activity was expressed as Trolox equivalents (TE) (μmol) per gram of sample based on the concentration required for 50% inhibition of DPPH coloration. 

The determination of total phenol in methanol extract by calorimetry is known as the method, Folin–Ciocalteu [42]. Firstly, a standard solution of gallic acid was prepared by dissolving in 80% of MeOH to obtain concentrations of 0.8, 0.4, 0.2, 0.1 and 0.05 mg/mL. Then, diluted Folin–Ciocalteu’s phenol reagent and 4% Na_2_CO_3_ solution were prepared. The measurement was conducted by using the 96-well crystal plates and a 96-well microplate reader (Infinite 200, Tecan, Japan, Tokyo). Plant crude extraction (10 μL), Folin–Ciocalteu’s phenol solution (75 μL) and 4% Na_2_CO_3_ solution were added to suitable wells. We measured the fluorescein by using the microplate reader at an absorbance of 765 nm. Results were expressed as mg gallic acid equivalent (GAE) per g of sample dry weight (DW) basis. 

### 4.3. Evaluation of Allelopathy of Plant Samples through Sandwich and Dishpack Method

In this experiment, the sandwich method was used to evaluate the allelopathic activity of plant extracts among Chinese pharmaceutical plants. This method was adopted from Fujii [43] and can measure a large number of samples efficiently in a short time. Agar medium (0.70%, *w/v*, autoclaved for 15 min at 115 °C) was used as the growth medium and lettuce was purchased as the plant receptor at the bio-assay, as their germination is very uniform and sensitive to allelochemicals [44]. To start with, 10 mg of plant samples were put into the multi-dish with 6 wells. For the next step, 10 mL agar (Nakalai Tesque Co., Ltd., Kyoto, Japan, Tokyo) was injected into the wells containing the plant samples. Last, 5 lettuce seeds (*Lactuca sativa* L., Takii Seed Co., Ltd., Japan, Tokyo) were carefully placed and the multi-dishes were wrapped in silver paper in dark conditions at 23 °C. After 3 days, the radicle and hypocotyl elongation of lettuce was determined. The plant growth elongation rate was calculated by measuring the radicle length of the sample, as shown in Equation (1). Agar with no addition of samples was set as control of untreated control. Data of treatment or untreated were described as the mean of the three replicates.
Elongation (%) = x/y × 100(1)
where x = mean value of sample radicle length, cm, and y = mean value of control radicle length, cm.

The allelopathy of volatile substances released from different parts of plants was screened by the bio-assay of dishpack, which can quickly identify the allelopathy characteristics and can be used for a large number of sample tests [45]. The 250 mg of collected samples were added to one of the wells in a six-well multi-dish plate. The 5 wells of the multi-well dish were covered with filter paper and filled with 0.7 ml of distilled water. After 7 lettuce seeds were placed on the filter paper of each well, the surface of the dishes was completely sealed with plastic tape to avoid the loss of volatile compounds. Immediately, dishes were placed into an incubator under completely dark conditions for 3 days at 22 °C. After 3 days, the extension of the radicle and hypocotyl was determined and recorded. Every bio-assay was replicated 3 times to obtain the mean value. No plant samples were added to the control treatment. 

### 4.4. Statistical Analysis

The assays of ORAC and DPPH-RSA and content of phenol were conducted in 3 duplicates and the original data were evaluated by mean (M) ± standard deviation (SD). Statistical analysis was performed using software of SPSS 13.0 and Excel 2003. The allelopathic activity was evaluated by calculating the means of radicle growth and the criterion of the SDV was calculated by using SPSS 13.0 and Microsoft Excel 2010. The radicle growth inhibition (% of untreated control) was described as 3 degrees: from the weakest to the strongest effects. (*) = M-0.5 SD, (**) = M-1.0 SD, (***) = M-1.5 SD.

## 5. Conclusions

The study aimed to estimate 55 Chinese pharmaceutical plants for antioxidative and allelopathic potential. The least radicle elongation of lettuce seedlings was caused by the fruit of sea buckthorn (*Hippophae rhamnoides*) at only 7.30 ± 0.35% of untreated control using the sandwich method. The fruit of sea buckthorn also a had high ORAC value (168 ± 7.04 μmol TE/g), DPPH radical scavenging activity (420 ± 7.15 μmol TE/g) and displayed a big amount of total phenol (236 ± 7.62 mg GAE/g). In addition, Star anise (*Illicium verum*) showed the strongest volatile allelopathic potential of 16.00 ± 0.14 (% of untreated control) by the dishpack method through volatilization. Moreover, star anise had higher levels at ORAC value (136 ± 4.20 μmol TE/g), DPPH-RSA (376 ± 5.04 μmol TE/g) and content of total phenolics (181 ± 6.20 mg GAE/g) than other volatile species. Lettuce growth inhibition by plant leachates had a strong significant positive correlation with antioxidant capacity. On the contrary, growth inhibition through volatile constituents had a weak significant positive correlation with antioxidant capacity. With this information, antioxidant capacity of plants can used as an indicator to select plants for screening for potential allelopathic species. A new hypothesis that antioxidative activity in plants is associated with allelopathic potential is proposed. Further studies using plants from other geographic locations are required. Studies on the isolation and identification of allelochemicals in candidate species from this research are ongoing.

## Figures and Tables

**Figure 1 plants-11-02481-f001:**
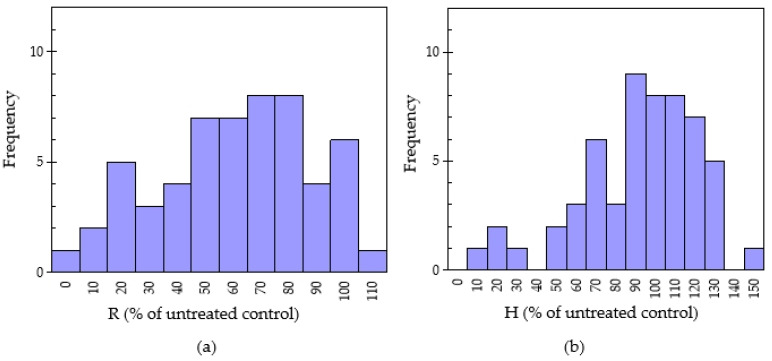
Normal distribution of radicle elongation (**a**) and hypocotyl elongation (**b**) of lettuce on 55 pharmaceutical plants treated by 10 mg sample using the sandwich method. R: Radicle and H: Hypocotyl.

**Figure 2 plants-11-02481-f002:**
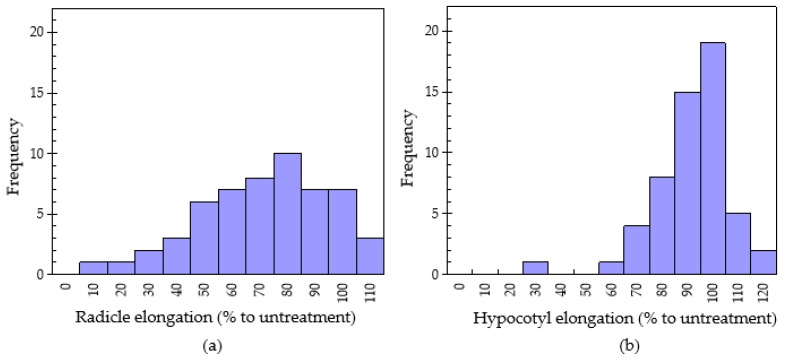
Normal distribution of radicle elongation (**a**) and hypocotyl elongation (**b**) of lettuce using dishpack method with 250 mg plant of the 55 medicinal plants.

**Figure 3 plants-11-02481-f003:**
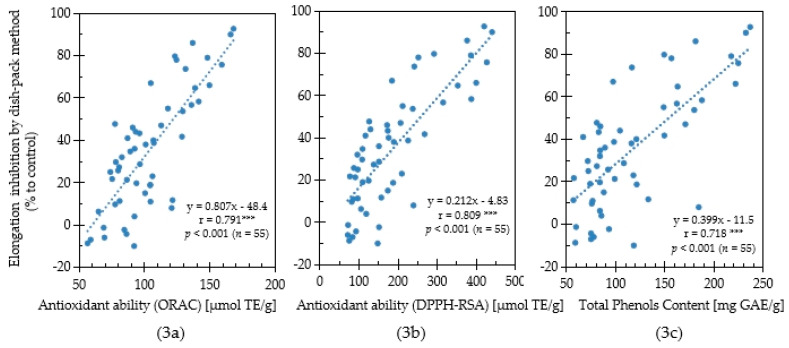
Correlation among ORAC antioxidant ability (**3a**), DPPH-RSA antioxidant ability (**3b**), total phenols content (**3c**) and lettuce elongation inhibition using the sandwich method for Chinese pharmaceutical plants. [***] indicates significant correlation when 0.5 ≤ |r| < 1.

**Figure 4 plants-11-02481-f004:**
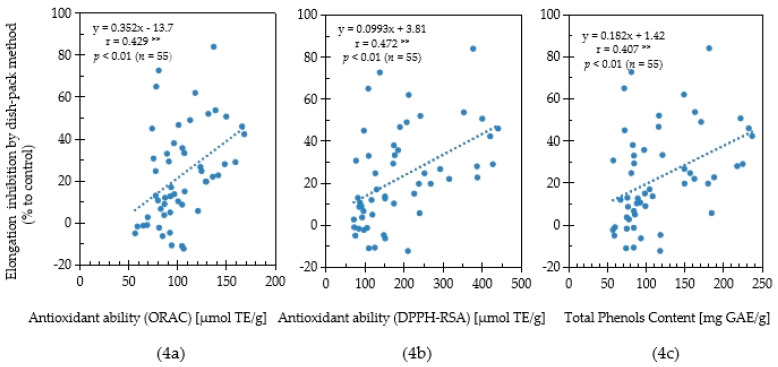
Correlation among ORAC antioxidant ability (**4a**), DPPH-RSA antioxidant ability (**4b**), total phenols content (**4c**) and lettuce elongation inhibition using the dishpack method for Chinese pharmaceutical plants. [**] indicates weak correlation when 0.3 ≤ |r| < 0.5.

**Table 1 plants-11-02481-t001:** ORAC, DPPH-RSA and total phenolic contents of 55 Chinese pharmaceutical plants.

Scientific Name	Plant Part Used	ORAC (μmol TE/g)	DPPH-RSA (μmol TE/g)	Total Phenolic Contents (mg GAE/g)
*Hippophae rhamnoides* L.	fruit	168 ± 7.04	420 ± 7.15	236 ± 7.62
*Hippophae rhamnoides* L.	leaf	166 ± 6.05	440 ± 7.32	232 ± 7.40
*Rosa multiflora* Thunb.	fruit	159 ± 5.38	426 ± 7.02	224 ± 7.51
*Punica granatum* L.	peel	149 ± 5.62	399 ± 6.40	221 ± 6.48
*Ribes nigrum* L.	fruit	148 ± 5.23	386 ± 5.17	217 ± 7.33
*Lycium ruthenicum* Murray	fruit	141 ± 5.01	387 ± 5.21	187 ± 6.04
*Capsicum annuum* L.	fruit	138 ± 5.21	352 ± 4.58	162 ± 4.01
*Illicium verum Hook.* f.	fruit	136 ± 4.20	376 ± 5.04	181 ± 6.20
*Physalis alkekengi* L.	fruit	135 ± 3.80	315 ± 4.20	162 ± 3.98
*Nitraria tangutorum* Bobr.	leaf	131 ± 4.05	241 ± 3.80	116 ± 2.75
*Rubus idaeus* L.	fruit	129 ± 3.52	237 ± 3.63	179 ± 5.31
*Cuminum cyminum* Linn.	fruit	125 ± 3.74	243 ± 2.15	162 ± 1.52
*Stachys geobombycis* C. Y. Wu	root	125 ± 2.10	249 ± 1.81	164 ± 2.30
*Gossypium herbaceum* L.	root	122 ± 2.15	290 ± 2.40	145 ± 2.13
*Adenia chevalieri* Gagnep	fruit	121 ± 3.22	157 ± 2.78	133 ± 3.44
*Dendrobium nobile* Lindl.	fruit	120 ± 3.10	239 ± 3.35	184 ± 4.86
*Indigofera tinctoria* L.	leaf	117 ± 2.18	211 ± 3.21	148 ± 3.67
*Schisandra chinensis* Baill.	fruit	112 ± 2.12	206 ± 3.10	170 ± 4.53
*Calophyllum inophyllum* L.	fruit	110 ± 1.67	208 ± 4.11	104 ± 2.25
*Perilla frutescens* Britt.	leaf	107 ± 0.94	217 ± 1.51	110 ± 1.40
*Ziziphus jujuba* Mill.	fruit	105 ± 0.51	200 ± 1.75	113 ± 1.30
*Chenopodium glaucum* L.	leaf	104 ± 1.73	184 ± 2.33	97.3 ± 2.07
*Heracleum scabridum* Franch.	root	104 ± 2.03	85.3 ± 1.60	76.4 ± 0.75
*Cicer arietinum* L.	fruit	104 ± 1.62	109 ± 2.46	74.0 ± 0.42
*Adenophora stricta* Miq.	fruit	104 ± 1.28	187 ± 2.35	121 ± 3.27
*Lonicera dasystyla* Rehd.	root	103 ± 2.30	185 ± 1.40	112 ± 1.80
*Paeonia lactiflora* Pall.	fruit	100 ± 1.37	173 ± 2.26	87.5 ± 2.05
*Aster tataricus* Linn.	leaf	98.4 ± 0.85	140 ± 1.60	93.0 ± 1.75
*Tribulus terrester* Linn.	fruit	95.4 ± 0.70	166 ± 3.01	102 ± 2.04
*Ziziphus jujuba* Mill.	fruit	93.8 ± 0.92	124 ± 1.52	83.5 ± 1.23
*Plantago asiatica* Linn.	leaf	93.7 ± 1.03	139 ± 1.30	94.7 ± 1.44
*Eriobotrya japonica* Lindl.	fruit	93.5 ± 2.41	151 ± 0.97	95.3 ± 1.37
*Munronia sinica* Diels	root	92.3 ± 0.68	118 ± 1.83	85.4 ± 1.90
*Morus macroura* Miq.	fruit	92.0 ± 0.77	147 ± 1.64	118 ± 2.90
*Aralia chinensis* L.	root	90.7 ± 0.55	171 ± 1.75	84.3 ± 1.82
*Elaeagnus angustifolia* Linn.	fruit	90.6 ± 1.2	100 ± 2.40	106 ± 1.51
*Dendranthema indicum* Moul	leaf	90.5 ± 2.10	110 ± 0.85	92.4 ± 1.32
*Morus alba* Linn.	root	88.2 ± 1.30	100 ± 0.82	73.4 ± 1.60
*Phyllanthus emblica* L.	fruit	86.5 ± 0.48	90.6 ± 1.21	98.7 ± 2.30
*Arnebia euchroma* Johnst.	leaf	86.0 ± 0.45	92.4 ± 1.03	74.8 ± 0.91
*Amaranthus viridis* L.	leaf	84.5 ± 0.37	151 ± 1.32	93.1 ± 2.13
*Armeniaca vulgaris* Lam.	fruit	82.7 ± 0.90	94.2 ± 1.41	87.5 ± 1.73
*Fritillaria walujewii* Regel	fruit	81.0 ± 0.27	96.7 ± 0.97	56.3 ± 0.26
*Angelica sinensis* Diels	root	80.7 ± 1.50	102 ± 2.10	83.4 ± 2.03
*Pueraria lobata* Ohwi	root	80.3 ± 1.73	90.6 ± 0.91	97.5 ± 4.23
*Rheum officinale* Baill.	root	79.3 ± 0.70	102 ± 0.91	79.3 ± 1.43
*Tulipa iliensis* Regel	leaf	77.4 ± 0.26	81.5 ± 0.80	75.6 ± 0.87
*Salix sinopurpurea* C. Wang	fruit	77.4 ± 0.22	126 ± 0.76	80.3 ± 1.04
*Trigonella foenum-graecum* L.	fruit	75.2 ± 0.30	76.9 ± 0.72	57.0 ± 0.41
*Dimocarpus longan* L.	fruit	75.0 ± 0.27	101 ± 0.51	80.0 ± 1.03
*Sinapis alba* L.	fruit	69.3 ± 0.13	71.3 ± 0.55	77.6 ± 0.95
*Senecio scandens* Buch.	fruit	68.7 ± 0.25	72.2 ± 0.64	59.6 ± 0.47
*Equisetum hyemale* L.	leaf	64.8 ± 0.14	105 ± 0.72	83.7 ± 1.64
*Armeniaca vulgaris* Lam	leaf	58.8 ± 0.17	83.7 ± 0.42	74.7 ± 0.80
*Carya cathayensis* Sarg	leaf	56.3 ± 0.12	75.1 ± 0.38	58.3 ± 0.30

Results are averages of three replications (M ± SD).

**Table 2 plants-11-02481-t002:** Radicle growth of 55 Chinese Pharmaceutical plants by sandwich method and dishpack method.

Scientific Name	Plant Part Used	Radicle Growth by Sandwich Method		Radicle Growth by Dishpack Method	
		(% to Control)	Criteria *	(% to Control)	Criteria *
*Hippophae rhamnoides* L.	fruit	7.30 ± 0.35	***	57.7 ± 2.51	*
*Hippophae rhamnoides* L.	leaf	10.0 ± 0.52	***	54.0 ± 1.75	*
*Illicium verum* Hook. f.	fruit	14.0 ± 0.48	***	16.0 ± 0.14	***
*Gossypium herbaceum* L.	root	19.3 ± 0.21	***	78.3 ± 1.03	
*Ribes nigrum* L.	fruit	21.0 ± 0.70	***	72.0 ± 4.38	
*Stachys geobombycis* C. Y. Wu	root	22.3 ± 0.65	***	75.3 ± 4.71	
*Rosa multiflora* Thunb.	fruit	24.3 ± 0.73	**	71.0 ± 4.13	
*Nitraria tangutorum* Bobr.	leaf	26.3 ± 0.78	**	48.0 ± 1.28	**
*Chenopodium glaucum* L.	leaf	33.0 ± 0.83	**	64.3 ± 2.30	*
*Punica granatum* L.	peel	34.0 ± 0.81	**	49.3 ± 1.86	**
*Capsicum annuum* L.	fruit	35.3 ± 0.90	**	46.3 ± 1.90	**
*Lycium ruthenicum* Murray	fruit	41.7 ± 0.95	*	77.3 ± 4.26	
*Physalis alkekengi* L.	fruit	43.3 ± 1.03	*	78.0 ± 3.97	
*Indigofera tinctoria* L.	leaf	45.0 ± 1.20	*	38.0 ± 0.82	***
*Rubus idaeus* L.	fruit	46.3 ± 1.08	*	80.3 ± 4.66	
*Salix sinopurpurea* C. Wang	leaf	52.3 ± 1.27		75.3 ± 5.10	
*Schisandra chinensis* Baill.	fruit	53.0 ± 1.30		51.0 ± 1.90	**
*Aralia chinensis* L.	root	54.0 ± 1.42		70.7 ± 4.40	
*Plantago asiatica* Linn.	leaf	56.3 ± 1.10		84.0 ± 2.10	
*Tribulus terrester* Linn.	fruit	57.7 ± 0.81		63.0 ± 2.03	*
*Cuminum cyminum* Linn.	fruit	58.7 ± 1.47		80.1 ± 4.11	
*Morus alba* Linn.	root	59.3 ± 2.03		82.3 ± 3.50	
*Perilla frutescens* Britt.	leaf	61.0 ± 3.70		65.7 ± 1.53	
*Calophyllum inophyllum* L.	fruit	61.7 ± 2.10		88.0 ± 5.42	
*Lonicera dasystyla* Rehd.	root	62.3 ± 2.70		56.1 ± 3.01	**
*Eriobotrya japonica* Lindl.	fruit	65.3 ± 1.80		89.0 ± 0.73	
*Elaeagnus angustifolia* Linn.	fruit	66.7 ± 2.70		65.0 ± 1.53	
*Dendranthema indicum* Moul.	fruit	67.0 ± 1.10		69.1 ± 1.81	
*Armeniaca vulgaris* Lam.	fruit	70.0 ± 2.30		98.5 ± 4.30	
*Rheum officinale* Baill.	root	70.7 ± 1.85		36.2 ± 1.13	***
*Aster tataricus* Linn.	leaf	71.0 ± 1.60		85.2 ± 3.23	
*Angelica sinensis* Diels	root	71.7 ± 3.24		29.1 ± 0.81	***
*Pueraria lobata* Ohwi	root	74.7 ± 2.51		87.3 ± 4.15	
*Dimocarpus longan* L.	fruit	76.7 ± 3.20		56.0 ± 1.70	*
*Solanum nigrum* Linn.	fruit	78.0 ± 4.05		112 ± 10.5	
*Trigonella foenum-graecum* L.	fruit	78.3 ± 5.20		69.3 ± 3.34	
*Phyllanthus emblica* L.	fruit	78.7 ± 5.35		91.0 ± 6.55	
*Ziziphus jujuba* Mill.	fruit	80.3 ± 6.24		110 ± 10.4	
*Adenophora stricta* Miq.	fruit	81.3 ± 7.40		58.3 ± 2.62	*
*Cicer arietinum* L.	fruit	81.0 ± 6.58		108 ± 9.22	
*Paeonia lactiflora* Pall.	fruit	85.0 ± 6.93		89.7 ± 6.90	
*Adenia chevalieri* Gagnep.	root	87.0 ± 7.56		69.0 ± 3.72	
*Fritillaria walujewii* Regel.	fruit	88.3 ± 8.03		102 ± 3.95	
*Heracleum scabridum* Franch.	root	88.7 ± 7.78		91.3 ± 7.31	
*Tulipa iliensis* Regel	root	89.0 ± 8.31		87.0 ± 6.70	
*Dendrobium nobile* Lindl.	fruit	90.3 ± 9.04		94.3 ± 8.11	
*Equisetum hyemale* L.	fruit	92.0 ± 8.61		101 ± 6.45	
*Munronia sinica* Diels	fruit	93.7 ± 9.30		95.0 ± 7.44	
*Senecio scandens* Buch.-Ham.	root	96.0 ± 9.22		101 ± 3.02	
*Amaranthus viridis* L.	fruit	101 ± 9.23		106 ± 9.63	
*Arnebia euchroma* Johnst.	leaf	102 ± 9.64		96.3 ± 7.80	
*Sinapis alba* L.	leaf	104 ± 9.70		97.3 ± 8.90	
*Armeniaca vulgaris* Lam.	fruit	106 ± 10.0		101 ± 8.86	
*Carya cathayensis* Sarg.	leaf	107 ± 9.85		105 ± 10.2	
*Morus macroura* Miq.	leaf	108 ± 10.10		104 ± 9.31	

R% = radicle elongation rate (% compared to untreated control) when applying the sandwich method or the dishpack method. More [*] indicates more excellent ability of radicle growth inhibition when the standard deviation variance (SDV) adopted, where: * = M-0.5 SD, ** = M-1.0 SD, *** = M-1.5 SD, M = Average of R%, SD = Standard deviation of R%.

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
