# Peer review of "Relationship between the Antioxidant Activity and Allelopathic Activities of 55 Chinese Pharmaceutical Plants"

_plants, 2022, doi:10.3390/plants11192481_

Round 1

Reviewer 1 Report

I have a lot of comments. See enclosed file, please.

Author Response

Cover Letter

Title: Relationship between the Antioxidant Activity and Allelopathic Activities of 55 Chinese Pharmaceutic Plants

Authors:  Aniya                         E-mail:  any2014@foxmail.com

          Xia Qile *                      E-mail:  cookxql@163.com  

          Yoshiharu Fujii *                E-mail:  yfujii@cc.tuat.ac.jp

          Yoshihiro Nomura              E-mail:  ny318@cc.tuat.ac.jp

          Kwame Sarpong Appiah         E-mail:  ksappih90@gmail.com

          Erdeng Fu                      E-mail:  398069845@qq.com

Yoko Suzuki                    E-mail:  yoko86252539@gmail.com

Manuscript ID: plants-1847879

Special Issue

New Insights into Plant Resistance to Stress

Dear editors and reviewers,

Thank you for giving us the opportunity to submit a revised draft of the manuscript “Relationship between the Antioxidant Activity and Allelopathic Activities of 55 Chinese Pharmaceutic Plants”. We appreciate reviewers effort to reviewing our manuscript and their positive feedbacks. The reviewer gives an accurate summary of our work and brings forward constructive questions. We have addressed them at the responses to the reviewers’ comments. We have incorporated most of the suggestions made by the reviewers. Those changes were revised using the function of “Track Changes” throughout the revised manuscript. Could editors and reviewers check the responses and revised manuscript in the system ?

Thank you and all the reviewers for the kind advice.

Sincerely yours,

Aniya.

Response to Reviewer 1 Comments

Question 1: What about capital and smal letters in the title?

Response 1: The representation of capital and smal letters in the title was revised. The content of “Relationship between the Antioxidant activity and Allelopathic Activities of 55 Chinese Medicinal Plants” was replaced by “Relationship between the Antioxidant Activity and Allelopathic Activities of 55 Chinese Pharmaceutic Plants”.

Question 2: Folin-Ciocaltenus?

Response 2: "Folin-Ciocaltenus" was replaced by "Folin-Ciocalteu" as showed in Abstract and other parts of manuscripts (Introduction, Results, Discussion, Materials and Methods and Conclusions)

Question 3: ha is hour (h) or something?

Response 3: “L/ha” was deleted and the content was revised. The content of “Manuka oil and its main component, leptospermone, inhibited the growth of 90% of the weeds tested at a dose of 3.1 and 1.1 L/ha” was replaced by “Macarook oil and its flavonoids can inhibit the growth of most weeds at a certain dose and maintain stability for 7 days”.

Question 4: mg/g or mg GAE/g?

Response 4: The original content of “Sea buckthorn evaluation using maceration, soxhlet, and subcritical water extraction techniques showed the antioxidant potential of 86.35, 133.31 - 255.87, and 164.03 - 343.86 Trolox equivalents per gram (TE/g), respectively, while as the respective total phenolic content was reported to be 28.35, 43.77 - 77.85, and 60.22 - 86.70 mg/g” was replaced by “Sea buckthorn was extracted by water extraction, soxhlet extraction and impregnation respectively and its antioxidant value was 164.03, 133.31 and 86.35 Trolox equivalents (TE) per gram respectively, while phenolics content was 60.22, 43.77 and 28.35 mg GAE/g mg/g respectively ”.

Question 5: What about mean particle size of samples?

Response 5: The content of “Plant samples were freeze-dried at -20°C for 24 h and were crushed into powder by a grinder ” was replaced by “After lyophilization at -20℃for 24 h, plant materials were crushed into powder by a grinder and filtered with a 100-hole sieve ”.

Question 6: "Plant materials were dissolved in 80% methanol to obtain the crude extracts."?

Response 6: The content of “ Plant materials were dissolved in 80% methanol to obtain the crude extracts. ” was replaced by “Plant materials were dissolved in 80% methanol to obtain the plant extract. ” 

Question 7: “where x = average of sample radicle length and y=average of control radicle length” Units?

Response 7: Units was added. The content of “Where x = average of sample radicle length and y=average of control radicle length” was replaced by “Where x = mean value of sample radicle length, cm and y= mean value of control radicle length, cm ”

Reviewer 2 Report

The manuscript entitled " Relationship between the antioxidant activity and allelopathic activities of 55 Chinese medicinal plants" explored the effect of the antioxidant activity on the allelopathic activity. The manuscript is quantitatively consistent but remains relatively poor in terms of quality. I think that it will be interesting to explore the phytochemical composition and determine the products responsible of the observed allelopathic/antixidant activities.

Other minor corrections are needed such as:

* Please replace "Folin-Ciocaltenus" by "Folin-Ciocalteu" throughout the manuscript

* Please italicize the scientific names of the explored plants.  

Author Response

Cover Letter

Title: Relationship between the Antioxidant Activity and Allelopathic Activities of 55 Chinese Pharmaceutic Plants

Authors:  Aniya                             E-mail:  any2014@foxmail.com

               Xia Qile *                         E-mail:  cookxql@163.com  

               Yoshiharu Fujii *             E-mail:  yfujii@cc.tuat.ac.jp

               Yoshihiro Nomura          E-mail:  ny318@cc.tuat.ac.jp

          Kwame Sarpong Appiah      E-mail:  ksappih90@gmail.com

                Erdeng Fu                        E-mail:  398069845@qq.com

               Yoko Suzuki                      E-mail:  yoko86252539@gmail.com

Manuscript ID: plants-1847879

Special Issue

New Insights into Plant Resistance to Stress

Dear editors and reviewers,

Thank you for giving us the opportunity to submit a revised draft of the manuscript “Relationship between the Antioxidant Activity and Allelopathic Activities of 55 Chinese Pharmaceutic Plants”. We appreciate reviewers effort to reviewing our manuscript and their positive feedbacks. The reviewer gives an accurate summary of our work and brings forward constructive questions. We have addressed them at the responses to the reviewers’ comments. We have incorporated most of the suggestions made by the reviewers. Those changes were revised using the function of “Track Changes” throughout the revised manuscript. Could editors and reviewers check the responses and revised manuscript in the system ?

Thank you and all the reviewers for the kind advice.

Sincerely yours,

Aniya.

Response to Reviewer 2 Comments

Question 1: I think that it will be interesting to explore the phytochemical composition and determine the products responsible of the observed allelopathic/antixidant activities.

Response 1: Thanks for the valuable advices from the reviewer. We agree that it will be interesting to explore the phytochemical composition and determine the products responsible of the observed allelopathic/antixidant activities and our group will do the relavent research later. In addition, it will be the time-consuming and money-consuming work because of the complex procedures for extracting phytochemicals. In fact, we can use the method by evaluation of specific activity and total activity to identify the allelochemicals after screening of a big amount of plant species with allelopathic potential by sand-wich or dish-pack method. The strength of allelochemicals can be assessed by the biological activity of a compound as expressed by EC50. The EC50 is the effective concentration of a compound to induce half of the maximum action. This activity is expressed by the specific concentration of the compound and is termed as “specific activity”. In this experiment, specific activity was determined based on the concentration of crude extract or pure compound and the percentage inhibition of plant growth. Compounds with a high specific activity can potentially be used as pesticides. Another term that characterizes allelochemical action is the “total activity”. The total activity of a compound is a function of its specific activity and its content in the plant and via this value, the contribution and influence of the compound at the allelopathic effect can be evaluated.

Question 2: Please replace "Folin-Ciocaltenus" by "Folin-Ciocalteu" throughout the manuscript

Response 2: "Folin-Ciocaltenus" has been replaced by " Folin-Ciocalteu" throughout the manuscript using the function of "Track Changes"

Question 3: Please italicize the scientific names of the explored plants

Response 3:  The scientific names of the explored plants have been italicized throughout the manuscript using the function of " Track Changes"

Reviewer 3 Report

I recommend the article to be published after major revision and after verifying the following comments:

1 / please underline the novelty of the work in the introduction

2 / please make a research hypothesis

3 / please verify the accuracy and correctness of Latin names; please remember to italicize

4 / units: umol we write together, not separately - please correct throughout the manuscript

5 / Folin method, this is a wrong definition - it is a spectrophotometric method with Folin-Ciocalteu reagent - please correct

6 / values ​​- it is unacceptable to use, e.g. 166 +/- 1.55 -> if a value is written with 2 decimal places, then the same should be written as an error - in this case it should be: 166.00 +/- 1.55

please revise throughout the manuscript

7 / Table 1 - no statistical analysis - please add homogeneous groups

8 / The discussion is very poor, please add publications, e.g. 10.1016/j.foodchem.2016.01.090; 10.1007/s00217-017-2884-4; 10.3390/antiox8120618; 10.1016/j.jfca.2021.104107 etc.

9 / Methodology - please standardize the record of the methods so that they are all written according to one scheme; once the information appears in what units the results are expressed in, sometimes it is missing; please verify where the subscripts and superscripts should be in the names

10 / Please provide for all devices used: name (model, manufacturer, country, city)

11 / please edit the entire manuscript in terms of style

12/ please expand the conclusions

13/ please add DOI in references

Author Response

Cover Letter

Title: Relationship between the Antioxidant Activity and Allelopathic Activities of 55 Chinese Pharmaceutic Plants

Authors:  Aniya                                E-mail:  any2014@foxmail.com

                Xia Qile *                           E-mail:  cookxql@163.com  

              Yoshiharu Fujii *                 E-mail:  yfujii@cc.tuat.ac.jp

             Yoshihiro Nomura               E-mail:  ny318@cc.tuat.ac.jp

          Kwame Sarpong Appiah        E-mail:  ksappih90@gmail.com

              Erdeng Fu                            E-mail:  398069845@qq.com

             Yoko Suzuki                         E-mail:  yoko86252539@gmail.com

Manuscript ID: plants-1847879

Special Issue

New Insights into Plant Resistance to Stress

Dear editors and reviewers,

Thank you for giving us the opportunity to submit a revised draft of the manuscript “Relationship between the Antioxidant Activity and Allelopathic Activities of 55 Chinese Pharmaceutic Plants”. We appreciate reviewers effort to reviewing our manuscript and their positive feedbacks. The reviewer gives an accurate summary of our work and brings forward constructive questions. We have addressed them at the responses to the reviewers’ comments. We have incorporated most of the suggestions made by the reviewers. Those changes were revised using the function of “Track Changes” throughout the revised manuscript. Could editors and reviewers check the responses and revised manuscript in the system ?

Thank you and all the reviewers for the kind advice.

Sincerely yours,

Aniya.

Response to Reviewer 3 Comments

Comment 1: please underline the novelty of the work in the introduction

Response 1: The description of novelty of the work has been added using the function of "Track Changes" in the revised manuscript. Please refer to the following content in the part of Introduction :

"In the Labiatae, Leguminosae and compositae of traditional Chinese medicinal plants, a large number of secondary metabolites formed by phenylpropanoids, including flavonoids, monophenols, lignans, phenolic acids and other phenolic substances, have been found......Understanding the relevance between allelopathy and antioxidation will help us to screen the large amount of raw materials with allelopathic potential efficiently."

Comment 2: please make a research hypothesis

Response 2: A research hypothesis has been added using the function of "Track changes in the revised manuscript. Please refer to the following content:

“In view of the dual role of phenolic substances in allelopathy and antioxidation, there was a hypothesis that antioxidative capacity had a positive correlation with allelopathic effect. Plant secondary metabolites including phenolic compounds or other antioxidative compounds through leaching released into organisms and may caused the accumulation of soluble bio-chemicals and the phenomenon of allelopathy.”

Comment 3: please verify the accuracy and correctness of Latin names; please remember to italicize

Response 3: Thanks for the valuable comment. It is important to correct the expression of scientific names. Latin names has been modified into Italicize using the function of “Track Changes” in the revised manuscript.

Comment 4: units: umol we write together, not separately - please correct throughout the manuscript

Response 4: Thanks for the kind comment. It is important to correct the expression of units. The expression of units has been verified using the function of "Track Changes" in the revised manuscript.

Comment 5: Folin method, this is a wrong definition - it is a spectrophotometric method with Folin-Ciocalteu reagent - please correct

Response 5: Thanks for the kind comment. Representation of Folin-Ciocalteu has been modified using the function of "Track Changes" in the revised manuscript.

Comment 6: values - it is unacceptable to use, e.g. 166 +/- 1.55 -> if a value is written with 2 decimal places, then the same should be written as an error - in this case it should be: 166.00 +/- 1.55

Response 6: In this study, we obeyed the rule of significant figures (significant digit). The reliability of the data was usually minimum in three digits, so all data from the manuscript was expressed as three digits, such as 167,12.3,0.123,0.0234 and was consistent throughout the manuscript.

Comment 7: Table 1 - no statistical analysis - please add homogeneous groups

Response 7: Data at Table 1 were  not taken at the same time and could not apply the common statistical analysis. Then we showed the results by the average of three replications and standard deviation(M±SD).

Comment 8: The discussion is very poor, please add publications, e.g. 10.1016/j.foodchem.2016.01.090; 10.1007/s00217-017-2884-4; 10.3390/antiox8120618; 10.1016/j.jfca.2021.104107 etc.

Response 8: The discussion has been revised and publications has been added using the function of "Track Changes" in the revised manuscript. Please refer to the part of discussion in the revised manuscript.

Comment 9: Methodology - please standardize the record of the methods so that they are all written according to one scheme; once the information appears in what units the results are expressed in, sometimes it is missing; please verify where the subscripts and superscripts should be in the names

Response 9: Thanks for the valuable comments and these comments and views were very helpful.The part of methodology has been verified and modified using the function of " Track Changes" in the revised manuscript, especially for the representation of units as well as the subscripts and superscripts.

Comment 10: Please provide for all devices used: name (model, manufacturer, country, city)

Response 10: All devices used: name (model, manufacturer, country, city) has been provided according to the requirements.

Comment 11: please edit the entire manuscript in terms of style

Response 11: The entire manuscript has been edited interms of style,especially for the font size, font style and paragraph distance.

Comment 12: please expand the conclusions

Response 12: The conclusions has been expand using the function of “Track Changes” in the revised manuscript. Please refer to the part of conclusions in the revised manuscript.

Comment 13: please add DOI in references

Response 13: DOI has been added in references using the functions of "Track Changes" in the revised manuscript.

Round 2

Reviewer 1 Report

It is OK for me now.

Author Response

New Comment from reviewer 1:It is OK for me now.

Response:

We appreciate the time you took to review this manuscript. The comments and suggestions provided in this review have helped to improve the quality of this manuscript.

Reviewer 2 Report

The reviewer highly appreciate the efforts made by the authors to enhance the scientific quality of the manuscript. The reviewer also agree with the authors reply concerning the compounds responsible for the observed antioxidant/allelopathic activities. The importance of such determination could be incorporated in the conclusion indicating that it will be done later.

Author Response

New Comment from reviewer 2:

The reviewer highly appreciate the efforts made by the authors to enhance the scientific quality of the manuscript. The reviewer also agree with the authors reply concerning the compounds responsible for the observed antioxidant/allelopathic activities. The importance of such determination could be incorporated in the conclusion indicating that it will be done later. 

Response 2:

We appreciate the kind advice and positive evaluation from the reviewer. The conclusion of the manuscript has been revised accordingly.

Reviewer 3 Report

In current form I recommend for publication in Plants.

Author Response

New Comment from reviewer 3:

In current form I recommend for publication in Plants.

Response 3:

We appreciate the time you took to review this manuscript. The comments and suggestions provided in this review have helped to improve the quality of this manuscript.
